# Modulating the Nature of Ionizable Lipids and Number of Layers in Hyaluronan-Decorated Lipid Nanoparticles for In Vitro Delivery of RNAi

**DOI:** 10.3390/pharmaceutics16040563

**Published:** 2024-04-20

**Authors:** Victor Passos Gibson, Houda Tahiri, Claudia Gilbert, Chun Yang, Quoc Thang Phan, Xavier Banquy, Pierre Hardy

**Affiliations:** 1Department of Pharmacology and Physiology, Université de Montréal, Montréal, QC H3T 1J4, Canada; victor.passos.gibson@umontreal.ca; 2Research Center of CHU Sainte-Justine, Université de Montréal, Montréal, QC H3T 1C5, Canada; houda.tahiri.hsj@ssss.gouv.qc.ca (H.T.); claudia.gilbert2.hsj@ssss.gouv.qc.ca (C.G.); chun.yang.hsj@ssss.gouv.qc.ca (C.Y.); 3Faculty of Pharmacy, Université de Montréal, Montréal, QC H3T 1J4, Canada; quoc.thang.phan@umontreal.ca (Q.T.P.); xavier.banquy@umontreal.ca (X.B.); 4Department of Pediatrics, Université de Montréal, Montréal, QC H3T 1J4, Canada

**Keywords:** lipid nanoparticles, layer-by-layer, siRNA, surface modification, glioblastoma

## Abstract

Lipid nanoparticles (LNPs) have established their position as nonviral vectors for gene therapy. Tremendous efforts have been made to modulate the properties of LNPs to unleash their full clinical potential. Among the strategies being pursued, the layer-by-layer (LbL) technique has gained considerable attention in the biomedical field. Illuminated by our previous work, here we investigate if the LbL approach could be used to modify the LNP cores formulated with three different ionizable lipids: DODMA, MC3, and DODAP. Additionally, we wondered if more than three layers could be loaded onto LNPs without disrupting their gene transfection ability. Taking advantage of physicochemical analysis, as well as uptake and gene silencing studies, we demonstrate the feasibility of modifying the surface of LNPs with the LbL assembly. Precisely, we successfully modified three different LNPs using the layer-by-layer strategy which abrogated luciferase activity in vitro. Additionally, we constructed a 5×-layered HA-LNP containing the MC3 ionizable lipid which outperformed the 3×-layered counterpart in transfecting miRNA-181-5p to the pediatric GBM cell line, as a proof-of-concept in vitro experiment. The method used herein has been proven reproducible, of easy modification to adapt to different ionizable lipid-containing LNPs, and holds great potential for the translation of RNA-based therapeutic strategies.

## 1. Introduction

Lipid nanoparticles (LNPs) for gene delivery have gained considerable attention in recent years with the ground-breaking clinical approval of Onpattro in 2018, and mRNA-1273 and BNT162b in 2020 [1,2,3,4]. The nonviral vectors have allowed the clinical translation of short interfering RNA (siRNA) and messenger RNA (mRNA) which face challenging physiological hurdles that prevent them to reach bedside by themselves [5]. Notwithstanding the important therapeutic landmarks enabled by LNPs, intense research is still underway to unleash the full potential of these vehicles. The demonstrated clinical translation of LNPs for gene delivery has paved the way for several investigations that aim at bringing to the clinics other RNA- and DNA-based cargos, such as gene editing tools, micro RNAs (miRNAs), and pDNA [6,7,8]. Another line of research focuses on the modulation of lipid components that constitute LNPs, with a special focus on the synthesis of novel lipid or lipoid structures for potent gene transfection in vitro and in vivo [9,10], or novel cationic lipids with the dual ability of gene transfection and responsiveness to stimuli [11,12,13]. Modulating the nature of the lipid components as well as changing the route of administration of LNPs are successful examples of extrahepatic delivery of payloads by LNPs [14,15,16]. A third pathway explores the surface modification of LNPs to tailor their targetability to specific cells or organs [17].

Here we focus on the surface modification of LNPs using the layer-by-layer (LbL) technique. The LbL strategy consists of depositing layers of polyelectrolytes of opposing charges on top of surfaces [18]. This technique has been largely used in material sciences and made its way to biomedical applications, providing therapeutically relevant properties to nanoparticles such as stage release of cargo, longer circulation time in vivo, targeted properties, and improved stability, the latter being an issue of utmost importance in the research of lipid-based nanocarriers [19,20,21,22,23,24,25,26]. By formulating LNPs containing the ionizable lipid DODMA for posterior modification with the LbL strategy, we demonstrated that hyaluronan-finished LNPs were able to deliver miR-181a to glioblastoma cells both in vitro and in vivo. miR-181a is implicated in the oncogenic process of several malignancies and strongly correlated to glioblastoma [27]. In our investigation, the encapsulation of miR-181a was achieved with high yield due to the washless method proposed therein and its local delivery induced strong tumor suppression in vivo [28].

To build the LbL assembly, we chose three specific polyelectrolytes: a layer of siRNA, to increase the loading cargo, poly-l-arginine, a polycation that could boost the endosomal escape of LbL LNPs, and hyaluronic acid, a natural target of CD44 receptors [29] that can direct the LNPs to CD44-overexpressing cancer cells [30].

In this investigation, we wondered if the LbL strategy can be expanded to LNPs formulated with different ionizable lipids, namely MC3 and DODAP. The ionizable lipid MC3 enabled the clinical translation of siRNA to the clinic. DODAP is one of the first ionizable lipids proposed for the siRNA transfection in vitro. Additionally, we investigated if more than three layers could be built onto LNPs formulated with MC3 or DODMA as ionizable lipids. To evaluate the effect of such modifications on LNPs, we assessed the physical–chemical properties of LNPs by DLS and by encapsulation efficiency, and we assessed the uptake of LbL-modified LNPs by confocal microscopy. Finally, we interrogated the gene transfection of the particles by delivering siRNA targeting the luciferase gene to the reporter luciferase-expressing glioblastoma cell line, GL261.

## 2. Materials and Methods

### 2.1. Materials

Solvents and chemical reagents were purchased from Sigma (Oakville, ON, Canada) and Thermo Scientific (Waltham, MA, USA). Cholesterol, 1,2-dioleyloxy-3-dimethylaminopropane (DODMA), 1,2-dioleoyl-sn-glycero-3-phosphoethanolamine (DOPE), and 1,2-dioleoyl-sn-glycero-3-phosphoethanolamine-N-(lissamine rhodamine B sulfonyl) (ammonium salt) (rhodamine-PE) were obtained from Avanti Polar Lipids (Alabaster, AL, USA). (6Z,9Z,28Z,31Z)-Heptatriaconta-6,9,28,31-tetraen-19-yl 4-(dimethylamino)butanoate (MC3) and 1,2-dioleoyl-3-dimethylammonium-propane (DODAP) were purchased from Nanosoft Polymers (Winstom-Salem, NC, USA). Poly-L-Arginine (5000–15,000 Da) was obtained from Sigma (Oakville, ON, Canada). Hyaluronic acid (60 KDa) was obtained from Lifecore Biomedical Inc. (Chaska, MN, USA). Oligonucleotides were obtained from Horizon Discovery (Waterbeach, UK). siRNAs were custom-made with the following sequences: scramble siRNA (siCTL, 5′-UAGCGACUAAACACAUCAAUU-3′), scramble siRNA Cyanine-3 (siRNA-Cy3, 5′-Cy3-UAGCGACUAAACACAUCAAUU-3′), and siRNA Luciferase (siLuc, 5’-GGUCUGAGCUCCUUGAUAAUU-3′). Hoechst 33,342 was purchased from Cayman Chemical (Ann Arbor, MI, USA).

### 2.2. Synthesis and Characterization of HA-LNPs

The synthesis of hyaluronan-decorated LNPs was carried out as reported previously [28]. Briefly, the LNP cores were synthesized by the ethanolic injection technique using hand pipetting with the following proportion: 50:39:11% mol, ionizable lipid, cholesterol, DOPE, 0.22 μmol total lipids, N/P 4 lipid:siRNA charge ratio, 3:1 volume ratio, sodium acetate buffer (10 mM, pH 4):ethanol. After 15 min, the LNPs were diluted to 300 μL with sodium acetate buffer. Then, LbL assembly was performed as described, with the following proportion: 14.1 μL of LNP suspension was mixed with 7 μL of siRNA at 20 μM, for DODMA and MC3, or 10 μL for DODAP LNPs (siRNA diluted in NaAc buffer). After 10 min, the PLA layer was built (1 μL of PLA at 2 mg mL^−1^ diluted in MilliQ water). For three-layered LNPs (HA-LNPs 3×), 5 μL of hyaluronic acid (5 mg mL^−1^ diluted in MilliQ water) was added to the LNP/siRNA/PLA particles. For the five-layered system (HA-LNPs 5×), siRNA (10 μL, 20 μM, diluted in NaAc) and PLA (1 μL, 2 mg mL^−1^, diluted in MilliQ water) were added before a final hyaluronic acid layer (5 μL, 5 mg mL^−1^, diluted in MilliQ water). The formulations were let standing still for 10 min at RT right after each layering step.

Hydrodynamic diameter, polydispersity index, and ζ-potential were measured at 20 °C with a Malvern Zetasizer Nano ZS instrument (Malvern, Worcestershire, UK). Samples in 10 mM sodium acetate buffer were diluted 25× in MilliQ water to a final volume of 700 μL. Size measurements were performed with a scattering angle of 173°, and Z-average (intensity) values were reported. The voltage for ζ-potential was set at 150 V. Measurements were performed in at least triplicate.

### 2.3. Encapsulation Efficiency

The encapsulation of siRNA within LNP cores was executed as described previously using SYBR Gold assay (Thermo Scientific, Waltham, MA, USA) [28]. The amount of free siRNA and LNP-loaded siRNA (LNPs disrupted in Triton X-100 0.2% in TE buffer) were determined against a standard curve prepared in TE and TE + Triton buffer. LNPs were diluted 40x in TE buffer. To measure the % of siRNA loaded within the layers, we opted for an indirect method where HA-LNP particles were diluted 80X in TE buffer and compared to the fluorescence of free siRNA at an equivalent concentration diluted in the same fashion. Fluorescence values were plotted against a standard curve, and encapsulation efficiency of siRNA within the layers of HA-LNPs was normalized to free siRNA and expressed as % of that value.

### 2.4. Confocal Microscopy

Murine-derived glioblastoma cell line stably expressing the luciferase reporter protein (GL261-Luc) was purchased from PerkinElmer (Waltham, MA, USA, catalog number BW134246) and cultured in Dulbecco’s modified Eagle’s medium (DMEM; Thermo Scientific, Waltham, MA, USA) supplemented with 10% FBS and 1% PenStrep (Wisent Inc, Saint-Jean Baptiste, QC, Canada). For microscopy studies, 50 k GL261 cells were seeded in a coverslip-containing 24-well plate 12 h prior to the experiment for adherence. The following day, HA-LNPs 3× or 5× were added to the cells at the desired final concentration. After overnight incubation, cells were fixed with paraformaldehyde 4%, washed thrice with PBS, and stained with Hoechst 33,342 at 2 μg mL^−1^ for 30 min, followed by PBS washing and mounting over glass slides for imaging with a Leica TCS SP8 microscope (Wetzlar, Germany). Pictures were analyzed on Fiji software (version 2.9.0/1.53t).

### 2.5. Gene Transfection

For gene silencing experiments, we performed the inverted transfection approach. An amount of 10 μL of LNPs, HA-LNPs 3×, or HA-LNPs 5× were added to an empty transparent 96-well plate at the desired concentration (*n* = 8), and 10,000 GL261 cells were seeded, followed by 5 min centrifugation at 500× *g*, RT. Plates were rocked briefly before incubation in a humidified incubator for 48 h. At the end of the experiment, we measured both viability and luciferase activity under the same conditions. In half the wells, 10 μL of WST-8 reagent (Abcam, Cambridge, UK) was added to assess cellular viability based on the mitochondrial activity of viable cells. After one hour, on the other half of the wells, 100 μL of Steady-Glo Luciferase Assay System substrate (Promega, Madison, WI, USA) was added. After gentle shaking and 5 min of incubation, we transferred the cellular–luciferase substrate mix to a white 96-well plate to measure luciferase activity. Readings were performed on the CLARIOstar plate reader (BMG Labtech, Ortenberg, Germany, WST-8 absorbance at 460 nm, luciferase signal was collected at 580–80 nm). Testing conditions were normalized to non-treated cells and expressed as % of NT cells. For the viability study on the pediatric glioblastoma cell line (SF188, Catalog number SCC282, Sigma, Oakville, ON, Canada), cells were cultured in Minimal Essential Media (MEM, Thermo Scientific, Waltham, MA, USA) supplemented with 10% FBS, 1% PenStrep (Wisent, QC, Canada), and 2 mM L-Glutamine (Gibco, Thermo Scientific, Waltham, MA, USA). Prior to the experiment, 3000 SF188 cells were seeded in a 96-well plate for overnight adherence. The next day, cells were treated with HA-LNPs 3×- or 5×-layered containing miR-181a-5p (custom synthesis, GE Healthcare Dharmacon, Inc., Lafayette, CA, USA, Active strand: 5′ 5′-P.A.A.C.A.U.U.C.A.A.C.G.C.U.G.U.C.G.G.U.G.A.G.U 3′; Passenger strand: 5′ U.C.A.C.C.G.A.C.A.G.C.G.U.U.G.A.A.U.G.U.U.U.U 3′). After 72 h, 10 μL of WST-8 reagent (Abcam, Cambridge, UK) was added to assess cellular viability. Readings were performed on the CLARIOstar plate reader (BMG Labtech, Ortenberg, Germany, WST-8 absorbance at 460 nm). Absorbance was normalized to scramble-treated cells.

### 2.6. Statistical Analysis

All experiments were repeated thrice, and data are expressed as mean ± standard deviation. Graphs were generated in Prism Software (GraphPad Software, La Jolla, CA, USA, Version 10.1.1). Statistical analysis was performed using two-tailed Student’s *t*-test or one-way ANOVA with post hoc comparison through Tukey’s test, and expressed as * (*p* < 0.05), ** (*p* < 0.01), *** (*p* < 0.001), or **** (*p* < 0.0001).

## 3. Results

### 3.1. Formulation of HA-LNPs with Different Ionizable Lipids

Our first goal aimed at developing three LNP cores containing one of three different ionizable lipids used in siRNA transfection: DODMA, MC3, and DODAP. LNPs were synthesized by the ethanolic injection technique, where a three-part aqueous buffer containing siRNA was manually mixed by rapid pipetting in a one-part ethanolic lipidic solution containing ionizable lipid, cholesterol, and DOPE (50:39:11 mol ratio). After dilution in acidic buffer, particles were further modified using the layer-by-layer technique and characterized by size, polydispersity index (PDI), and ζ-potential (Figure 1A,B). Notably, DODAP required more siRNA in the first layering step than DODMA and MC3 LNPs. After the deposition of three polyelectrolytes (siRNA, poly-l-arginine, and hyaluronic acid) on top of LNPs, all particles increased in size. MC3-core LNPs increased from 120 ± 1 to 155 ± 1 nm (PDI of 0.21 ± 0.01 and 0.16 ± 0.01 for MC3 LNPs before and after LbL, respectively), DODMA LNPs increased from 146 ± 1 to 177 ± 2 (PDI of 0.27 ± 0.01 and 0.18 ± 0.02 for DODMA LNPs before and after LbL, respectively), and DODAP LNPs increased from 148 ± 2 to 174 ± 1 (PDI of 0.2 ± 0.01 and 0.18 ± 0.01 for DODAP LNPs before and after LbL, respectively). The correlogram for all three LNPs and HA-LNPs indicates good quality of the data, and the intensity distribution agrees with the PDI, which remained below 0.2 throughout the conditions for all three lipids. The ζ-potential (Figure 1B) alternates between positive and negative values after the addition of each polyelectrolyte, adding evidence to the successful layer-by-layer assembly of LNPs and each subsequent layer thereof. Before LbL modification, we quantified the encapsulation efficiency (EE%) of the three LNP cores using SYBR Gold intercalating reagent prior to and after disruption with triton 0.2%. The released siRNA was quantified against a standard curve. Core-loaded siRNA was encapsulated at a high percentage for all LNPs, with DODMA and MC3 being the highest (90 ± 1 and 89 ± 1, respectively), followed by DODAP (86 ± 1). Figure 1C depicts the encapsulation of layered siRNA for the three systems after LbL modification. For the layered siRNA, our strategy relied on measuring the fluorescence of free siRNA at a concentration equivalent to that added as the first polyelectrolyte layer deposited onto LNPs. The free siRNA solution was prepared using the same proportion of acidic buffer as HA-LNPs and diluted 80X in TE buffer. Then, we incubated HA-LNPs diluted at the same ratio with SYBR Gold 10X. The fluorescence detected in the HA-LNP samples were normalized against free siRNA and yielded a % of EE of 88 ± 1, 88 ± 1 and 86 ± 1, for DODMA, MC3, and DODAP LNPs, respectively. Finally, we incubated DODMA, MC3, and DODAP HA-LNPs, containing siRNA targeting the *firefly luciferase* gene (siLuc both in the core and first layer) with GL261-Luc cells to assess the in vitro transfection by measuring the luciferase activity of cells 48 h post-transfection (Figure 1D). All three systems were able to mediate protein silencing without eliciting significant cellular death at siLuc or siCTL concentrations of 180 nM. Here, the HA-LNP MC3 core induces the strongest gene silencing with final % luciferase activity of 25 ± 1, followed by 44 ± 2 for DODMA HA-LNPs, and 49 ± 1 for DODAP HA-LNPs.

### 3.2. Construction of Five-Layered Hyaluronan-Decorated LNPs (HA-LNPs 5×)

We then proceeded to answer our second question, concerning the deposition of five layers on top of LNPs (Figure 2).

We carried out this experiment with DODMA LNPs and MC3 LNPs only. The construction of polyelectrolyte layers followed a similar fashion than for the three-layered HA-LNPs, with the exception of the third layer (siRNA), where 10 µL of siRNA at 20 µM in 10 mM sodium acetate buffer (pH 4) was added to fully invert the ζ-potential. The correlogram and intensity distribution of non-modified LNPs and five-layered hyaluronan-decorated LNPs (HA-LNPs 5×) are depicted in Figure 3. Detailed size distribution, ζ-potential, and PDI after each layer are specified in Figure 4 for MC3-containing LNPs and Appendix A for DODMA-containing LNPs. Again, particles increased in a similar fashion than for HA-LNPs 3×, as can be observed in the right shift of intensity distribution for DODMA and MC3 HA-LNPs as compared to plain LNPs (Figure 2). The MC3 core increased from 110 ± 1 o 185 ± 2, and the DODMA core from 119 ±1 to 178 ± 1. PDI remained below 0.2 throughout all layer-by-layer steps for DODMA LNPs but increased to 0.3 in MC3 LNPs after the third layer (siRNA) and continued at 0.3 after the fourth (PLA) and terminal fifth (hyaluronic acid) layer.

The five-layered system also demonstrated high encapsulation efficiency of siRNA located at the first and third layers (Appendix A). Additionally, confocal microscopy confirmed the internalization of LNP cores modified with three or five layers (Appendix A). Similarly, siRNA-Cyanine 3 (siRNA-Cy3) located either in the first or third layers of HA-LNPs 5× were internalized by GL261 glioblastoma cells (Appendix A). The internalization was translated into gene transfection for both DODMA and MC3 HA-LNPs 5× (Figure 5). Luciferase silencing in HA-LNPs 5× was achieved at similar levels than HA-LNPs 3×, despite the location of siLuc in the first (bars 2 and 5, left to right, Figure 5) or third layer (bars 3 and 6, left to right, Figure 5). A small decrease in gene silencing is observed when siLuc is layered third on top of LNPs. This decrease is not significant in DODMA HA-LNPs 5×, but it is in MC3 HA-LNPs 5×. The trend was observed over multiple replicates of transfection of HA-LNPs 5× for both MC3 and DODMA cores.

We were then curious to understand if the slight decrease in gene silencing was to happen in HA-LNPs 5× compared to 3× if the siLuc was located only at the core of LNPs (Figure 6). We normalized the concentration of siLuc at 60 nM to add the same proportion of particles per well. No statistical difference was observed between MC3 HA-LNPs 3× and 5×, although the opposite trend was observed (the mean of MC3-core HA-LNPs 5× was numerically lower than 3×). Both conditions mediated strong gene silencing compared to non-modified MC3-core LNPs (MC3 HA-LNPs *p* < 0.001, and MC3 HA-LNPs 5× *p* < 0.0001, both compared to MC3-core LNPs), confirming that the layer-by-layer technique is necessary to stabilize non-PEGylated LNP cores. Interestingly, layer components only (siLuc, PLA, and HA) do not elicit gene silencing (Appendix A).

Taken together, our results attest to the reproducibility and feasibility of employing the layer-by-layer technique to modify lipid nanoparticles with different ionizable lipid components. Furthermore, increasing the number of layers maintains the internalization and gene transfection ability of LNPs, regardless of the location of therapeutic siRNA.

In order to test the transfection efficiency of layer-by-layer assembled LNPs in a more pharmacologically relevant model, we investigated the effect of micro-RNA-181a-5p (miR181a) on the viability of pediatric glioblastoma cells in vitro (SF188). We have previously demonstrated that miR181a is highly effective against adult GBM cells in vitro (U251, T98G, and U87 cells) and in vivo (U87 xenograft model) [28]. Here, our goal was two-fold: firstly, we wanted to evaluate if miR181a could be a potential target to treat pediatric GBM cells and, secondly, if any difference could be observed between the 3×- and 5×-layered HA-LNPs synthesized using the MC3 ionizable lipid as main component (Figure 7).

It can be observed that 5×-layered LNPs (MC3 core) performed slightly better than the 3×-layered HA-LNPs, but no statistical difference was observed if miR181a was located either in the 1st or 3rd layer of 5× HA-LNPs. Additionally, no toxicity was observed when a scramble miRNA was used for transfection, corroborating the safety of the 3×- and 5×-layered system, as observed when scramble siRNA was encapsulated within HA-LNPs for transfection in GL261-Luc cells. Further experiments will indicate if the 5×-layered system is more stable after intravenous administration in mice than the 3×-layered system.

## 4. Discussion

In this investigation, we extended the application of the layer-by-layer technique to modify lipid nanoparticles. Initially, we wondered if the reported methodology used first to modify LNPs containing DODMA as an ionizable lipid [28] could be reproduced in other LNPs with different ionizable lipid components. Interesting observations were drawn from this experiment. Our methodology was based on a titration method to identify the minimum amount of polyelectrolyte necessary to fully invert the ζ-potential without eliciting destabilization/aggregation of the system (two-fold increase in particle size and PDI) [28]. The method reported previously for DODMA LNPs was successfully reproduced to coat MC3 LNPs but not DODAP LNPs. For the layer-by-layer assembly, the first layer consisted of mixing 14.1 µL of LNPs at 733 µM (total lipid content) with 7 µL of siRNA at 20 µM. This proportion was enough to invert the charge of DODMA and MC3 LNPs without destabilization of the system (Figure 4, MC3 LNPs, and Appendix A, DODMA LNPs) but induced significant aggregation of DODAP LNPs with particles starting at 145 nm increasing over 1 µm. Therefore, a higher concentration of siRNA (10 µL at 20 µM) was necessary to invert the ζ-potential to a negative value able to maintain the colloidal stability of the system. Interestingly, there was no need to modify the proportion of PLA and HA as second and third layers, respectively (Figure 1). The higher ζ-potential of DODAP LNPs (ζ-potential > 30 mV) when compared to DODMA and MC3 LNPs (ζ-potential between 10 and 20) in sodium acetate buffer (10 mM, pH 4) may be an important parameter to take into account when optimizing the proportion of polyelectrolytes to be used in the LbL assembly of LNPs.

Quantifying the layered–encapsulated siRNA after the layer-by-layer strategy imposes its own challenge. Authors have successfully incubated the LbL system with large negatively charged polyelectrolytes which induce the release of electrostatic-bound layer components by competition [21]. We have tried such an approach with heparin, but we could not recover 100% of the encapsulated siRNA, probably due to the lower sensitivity of SYBR Gold compared to RiboGreen, as well as the fact that the first layer siRNA is bound to the surface of positively charged LNPs leading to complex structures that may protect the siRNA beyond solely electrostatic interactions [31]. Further characterization will elucidate how the LbL process shapes the structure of initial LNP cores. Therefore, we opted for an indirect quantification, where the equivalent amount of layered–encapsulated siRNA was incubated with the intercalating agent to have 100% fluorescence followed by incubation of HA-LNPs in the same fashion to determine the amount of free siRNA in solution. As demonstrated before, layered siRNA is well protected after the layer-by-layer technique, with less than 15% of siRNA detected in the three hyaluronan-decorated LNPs under investigation when compared to the fluorescence of free siRNA. We highlight that SYBR Gold might be able to detect the complexed siRNA by penetrating within layers giving room to the possibility that not all fluorescence detected comes from free siRNA in the solution. Again, the low fluorescence intensity detected for the SYBR Gold–siRNA complex corroborates the titration method used where we found the minimum amount polyelectrolyte to fully invert the ζ-potential of particles, thus avoiding an excess of free siRNA. Finally, all systems were able to mediate strong gene silencing when siRNA targeting the reporter protein luciferase was incorporated as the first layer. Not surprisingly, MC3 had the best performance compared to DODMA and DODAP HA-LNPs (Figure 1D). MC3 is one of the most used ionizable lipids for siRNA transfection and a core component of the first-ever siRNA-based therapeutics to reach the market [32]. Based on the highest transfection efficiency of MC3 and our previous work on DODMA LNPs, we decided to move both systems forward to answer our next question: could we build more than three layers—here we are investigating five—on top of LNPs?

To formulate the five-layered system, we started with LNP cores followed by siRNA and PLA layers as established for DODMA and MC3 LNPs, but the third layer was increased to 10 µL of siRNA at 20 µM as that was the best proportion to invert the charge of the particles. This indeed was sufficient to invert the surface charge and maintain the stability of the system. In fact, both systems were obtained below 180 nm size with ZP < −15 mV, the latter giving evidence of the presence of hyaluronic acid in the outermost layer. Again, the indirect quantification of siRNA did not unveil the presence of free nucleic acid (Appendix A). Interestingly, siRNA located either on the first or third layer is well-internalized in vitro (Appendix A). Encapsulating siRNA in the core of either HA-LNPs 3× or 5× also maintains gene transfection. The possibility of encapsulating siRNA in three different locations (core, first, or third layers) within the same particle could be an asset when one envisions a therapeutic approach with different RNAs, and when the high loading of genes per particle is desirable.

Titrating the amount of polyelectrolytes to increase the number of layers on top of soft materials, such as liposomes, has been successfully achieved before [33]. Authors have demonstrated that the concentration of the layering buffer, as well as the concentration of the polyelectrolyte to avoid aggregation due to excess materials, are important to maintain the stability and controlled assembly of the system. The influence of the ionic concentration of the buffer, the chemical nature of polyelectrolytes, and the template are examples of factors of utmost importance when building layer-by-layer assembled systems [34]. In this investigation, we maintained a low ionic strength buffer (10 mM sodium acetate) and the required concentration of polyelectrolytes at a minimum to have control over particle size and homogeneity. Through diligent in vivo experiments, Poon and coworkers have discussed that nanoparticles with an increased number of layers have longer circulation times than their single-layered counterparts [35]. In this investigation, gene transfection data with both siRNA targeting the luciferase gene and delivery of miR181a to pediatric GBM cells indicates the 5×-layered HA-LNPs system performed slightly better than the 3×-layered counterpart (Figure 4 and Figure 7). A few hypotheses could be drawn to explain the mechanism whereby the 5×-layered system outperformed the 3× HA-LNPs. First, the 5× HA-LNPs possess the double amount of Poly-L-Arginine as the 3× HA-LNPs. PLA is a known endosomal escape-inducing polypeptide that provides siRNA gene transfection ability to LbL-assembled nanoparticles [36]. Secondly, the likely higher stability of a multilayered system could also have an impact in the gene transfection in vitro. Further studies will investigate the role of PLA in the endosomal escape of HA-LNPs and the in vivo behavior of 3×-layered and 5×-layered HA-LNPs.

The layer-by-layer modification addresses some bottlenecks in the LNPs’ translation to the bedside. Hyaluronan-terminated nanoparticles are prone to less liver accumulation, a much-envisioned feature in the field of gene therapy using non-viral vectors. Additionally, the antifouling properties of hyaluronic acid might be of interest in the field where PEG-replacing strategies have been discussed [37,38,39,40]. These advantages can be expanded to other polyanions, as the outermost layer plays an important role in the intracellular trafficking of nanoparticles and their in vivo biodistribution [41,42]. Finally, ongoing research has been demonstrating the feasibility of developing LbL or their closely related core–shell-assemble nanoparticles by microfluidics [43,44], which is a technique that could be scaled up for industrial applications. Therefore, this investigation sheds light on the applicability of the LbL strategy to modulate the properties of LNPs, the gold standard gene delivery platform to date. Here, we demonstrate the feasibility of constructing multi-layered LNPs where both the number of layers and the nature of LNP cores can be modified. Further experiments will address the behavior of 3×- and 5×-layered systems on the fate and gene transfection of LNPs in vivo.

## 5. Conclusions

In summary, this study expands the data on the applicability of the layer-by-layer technique to lipid nanoparticles. We demonstrate that the methodology can be employed on LNPs formulated with different ionizable lipid cores which opens up a pathway where one can modulate the nature of lipids best suitable to the targeted gene to be delivered. The layer-by-layer strategy was necessary to stabilize non-PEGylated LNPs and allow gene transfection in vitro. Such observation confirms the position of depositing polyelectrolytes of alternative layers on top of LNPs to engineer effective non-viral vectors with the possibility of easy modulation of particles’ properties. Finally, we demonstrate the feasibility of depositing five layers on LNPs, where gene transfection in vitro is maintained despite the location of targeted siRNA. This work illustrates the benefits of using well-known techniques to modify the most clinically relevant gene delivery to date. The promising data gathered herein encourage us to understand and fully depict the physicochemical properties of LbL LNPs as well as explore their full potential by carefully choosing the nature of LNP components and polyelectrolytes.

## 6. Patents

The authors declare the following conflict of interest: Pierre Hardy, Victor Passos Gibson, Houda Tahiri, Chun Yang, and Xavier Banquy are listed as inventors in the US Patent application #N 63/4s8,667.

## Figures and Tables

**Figure 1 pharmaceutics-16-00563-f001:**
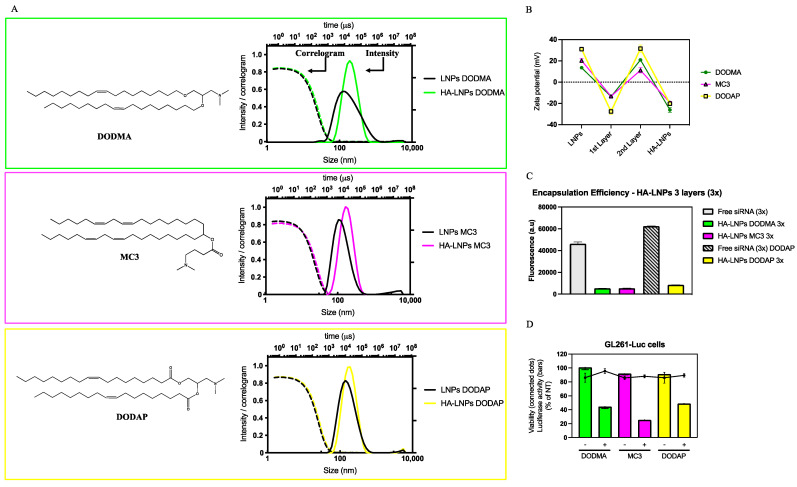
Characterization and in vitro silencing of hyaluronan-decorated LNPs (HA-LNPs) formulated with three different ionizable lipids (DODMA, green, MC3, magenta, DODAP, yellow). (**A**) Structure of the three different ionizable lipids used to formulate LNPs. Top to bottom: DODMA, MC3, and DODAP. Next to each structure, intensity size distribution and correlogram for LNPs before LbL modification (dashed line, black) and after the addition of the three polyelectrolytes to yield HA-LNPs (filled lines, color correspondent to each ionizable lipid used). (**B**) ζ-potential before and after each polyelectrolyte layer. (**C**) Encapsulation efficiency. (**D**) Luciferase silencing 48 h after transfection of three different HA-LNPs containing siRNA Luciferase both in the core (N/P 4) and layer at a final concentration of 180 nM of siRNA/well in GL261 cells stably expressing firefly luciferase gene. Scramble siRNA was used as a negative control and results were normalized to non-treated cells (luciferase activity, filled bars, viability, connected dots). All experiments were repeated at least three times.

**Figure 2 pharmaceutics-16-00563-f002:**
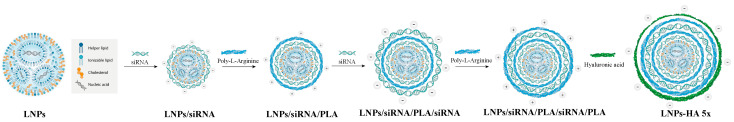
Schematic representation of five-layered hyaluronan-decorated lipid nanoparticles (HA-LNPs). Starting with positively charged LNPs, a layer of negatively charged siRNA was deposited onto the surface of LNPs, followed by Poly-L-Arginine (PLA). The third and fourth layer were constituted of siRNA and PLA, respectively. The fifth and final layer comprised hyaluronic acid.

**Figure 3 pharmaceutics-16-00563-f003:**
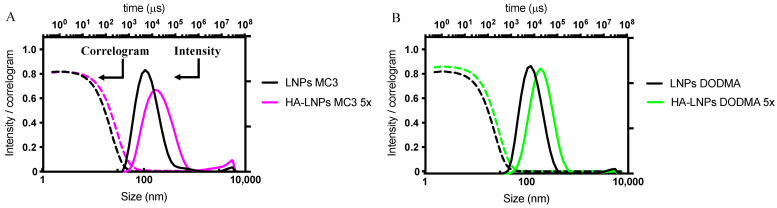
Intensity distribution and correlogram of LNPs formulated with either DODMA or MC3 and modified with five layers of polyelectrolytes (HA-LNPs 5×). Intensity and correlogram of initial LNPs (black solid and dashed lines, respectively) before and after the fifth-layer deposition with hyaluronic acid, to afford either (**A**) MC3 HA-LNPs 5× (magenta solid and dashed lines for intensity and correlogram, respectively) or (**B**) DODMA-formulated LNPs (green solid and dashed lines for intensity and correlogram, respectively).

**Figure 4 pharmaceutics-16-00563-f004:**
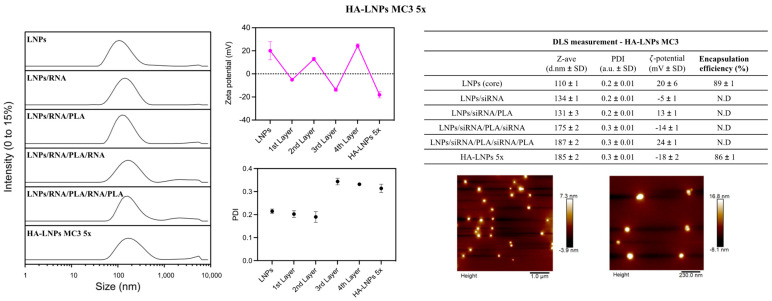
Physicochemical characterization of five-layered MC3-core HA-LNPs. (**Left**) Intensity distribution of MC3-core LNPs before and after each polyelectrolyte layer. (**Middle**) Zeta potential (middle top, in magenta) and PDI (dot plot, black) of MC3-core LNPs before and after each polyelectrolyte layer. (**Right**) Top table represents the mean and SD of z-average, PDI, and zeta potential at each step of the layer-by-layer technique. Encapsulation efficiency was determined for MC3-core LNPs with and without triton disruption of vesicles (0.2%) and HA-LNPs 5× using SYBR Gold as intercalating agent. For HA-LNPs 5×, a solution of free siRNA was used for 100% fluorescence control. (**Right**) Bottom pictures, AFM of MC3-core HA-LNPs 5×.

**Figure 5 pharmaceutics-16-00563-f005:**
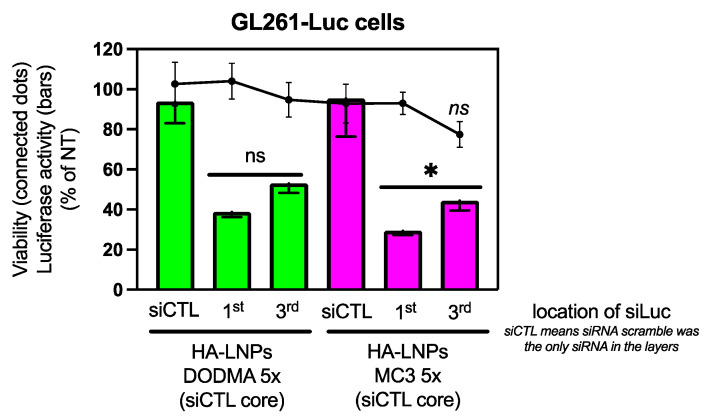
Luciferase silencing in GL261-Luc cells (180 nM, 48 h) mediated by siRNA-Luc encapsulated in the five-layered HA-LNPs formulated with either DODMA or MC3 ionizable lipid. Five-layered HA-LNPs formulated with either DODMA (green bars) or MC3 (magenta bars) were used to transfect siRNA Luciferase located at either the first or third layers, as indicated by the *x*-axis labelling (1st or 3rd). Control particles were formulated with scramble siRNA located in both 1st and 3rd layers of the 5× HA-LNPs (siCTL in the *x*-axis). The labeling 1st or 3rd indicate where the siRNA Luciferase is located. HA-LNPs containing siLuc in the 1st layer and siCTL in the 3rd layer are named 1st in the graph. HA-LNPs containing siCTL in the first layer and siLuc in the 3rd layer are named 3rd in the graph. Scramble siRNA (siCTL) was used in the core of all HA-LNPs. Two-way ANOVA multiple comparisons was used as a statistical tool (* *p* < 0.05, luciferase activity of MC3-core HA-LNPs 5× between siLuc located at the first layer versus MC3-core HA-LNPs 5× containing siLuc at the third layer; ns *p* > 0.05, viability difference between MC3-core HA-LNPs 5× containing siLuc at the third layer versus MC3-core HA-LNPs 5× containing siCTL at both first and third layers).

**Figure 6 pharmaceutics-16-00563-f006:**
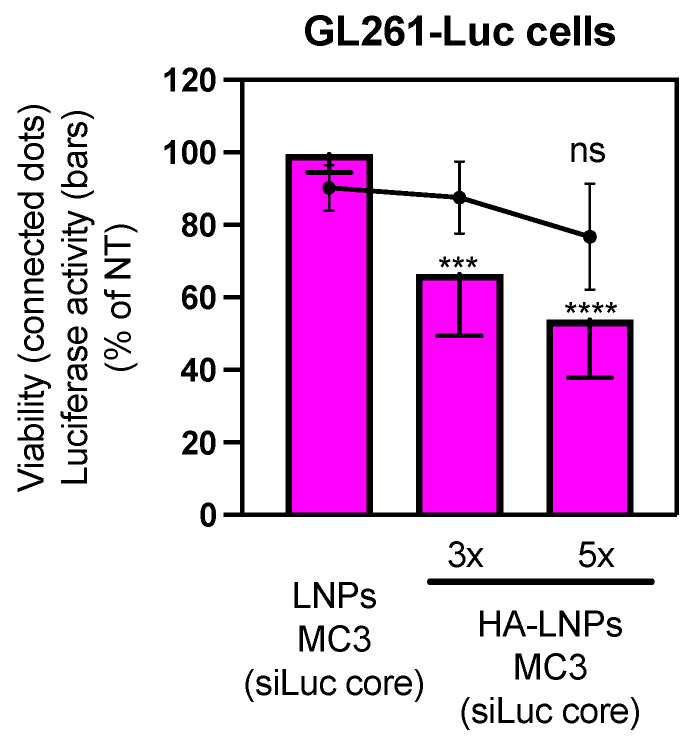
Luciferase silencing in GL261-Luc cells (60 nM, 48 h) mediated by siRNA-Luc encapsulated solely in the core of MC3 LNPs and hyaluronan-decorated 3×- and 5×-layered HA-LNPs. Filled bars and connected dots represent luciferase activity and viability compared to non-treated cells, respectively (not represented in the graph). The concentration was normalized across all conditions to 60 nM of siLuc per well. Two-way ANOVA multiple comparisons was used as a statistical tool (*** *p* < 0.001, **** *p* < 0.0001, luciferase activity of HA-LNPs 3× and HA-LNPs 5× compared to non-modified LNP cores, respectively; ns *p* > 0.05, viability difference between MC3-core HA-LNPs 5× and non-modified MC3-core LNPs).

**Figure 7 pharmaceutics-16-00563-f007:**
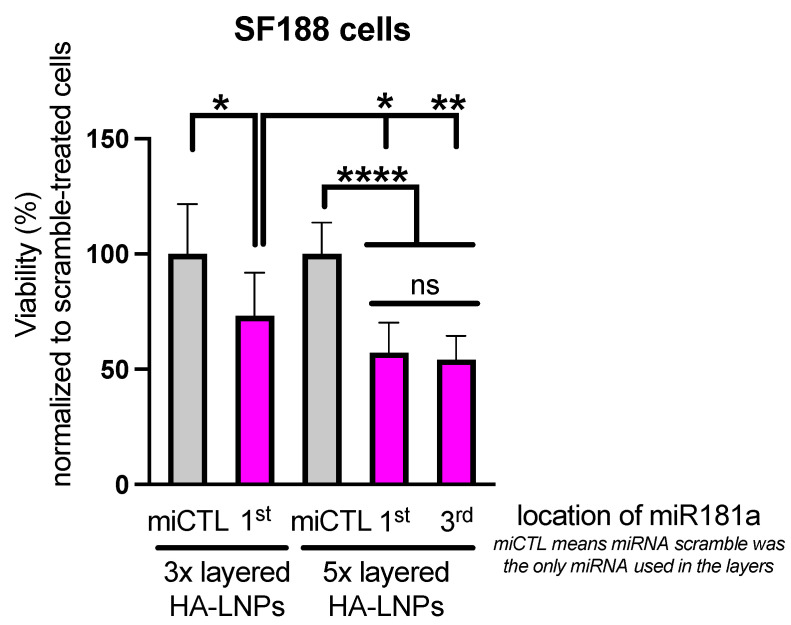
The effect of miR181a (142 nM) on pediatric glioblastoma cells SF188 72 h after transfection with either 3×- or 5×-layered HA-LNPs (MC3 core). Viability was measured using the WST-8 assay and values were normalized to scramble miRNA-treated cells. For the 5×-layered HA-LNPs, miR181a was either located in the 1st or 3rd layer and scramble miRNA was used to complete the layer-by-layer assembly of LNPs. All LNPs were formulated with scramble miRNA in the core compartment (N/P 4). Multiple comparisons were performed using two-way ANOVA (* *p* < 0.05, viability difference between the groups HA-LNPs 3× miCTL and HA-LNPs 3× containing miR-181a in the first layer; HA-LNPs 3× containing miR-181a in the first layer and HA-LNPs 5× containing miR-181a in the first layer; ** *p* < 0.001, viability difference between HA-LNPs 3× containing miR-181 in the first layer and HA-LNPs 5× containing miR-181a in the third layer; **** *p* < 0.0001, viability difference between HA-LNPs 5× containing miCTL and both HA-LNPs 5× containing miR-181a in the first and HA-LNPs 5× containing miR-181a in the third layer; ns *p* > 0.05, no statistical difference between HA-LNPs 5× containing miR-181a in the first layer and HA-LNPs 5× containing miR-181a in the third layer). Experiments were performed in triplicates with *n* = 8 wells per condition.

## Data Availability

The original contributions presented in the study are included in the article/Appendix A; further inquiries can be directed to the corresponding authors.

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
