# Peer review of "Modulating the Nature of Ionizable Lipids and Number of Layers in Hyaluronan-Decorated Lipid Nanoparticles for In Vitro Delivery of RNAi"

_pharmaceutics, 2024, doi:10.3390/pharmaceutics16040563_

Round 1

Reviewer 1 Report

Comments and Suggestions for Authors

The paper is well written and very interesting. I would propose only few editorial changes. 

1. Please explain the LbL abbreviation in the first use in abstract.

2. I would suggest to move some of the figures, at least 1 and 3, 5 and 6 from the Supplementary materials to the manuscript.

3. Please explain how the precision of indirect measurement of loading  efficiency was established?

Reviewer 2 Report

Comments and Suggestions for Authors

Pierre Hardy et al. reported an interesting study on siRNA delivery by LNP. The topic is a current hotspot in the area of pharmaceutics and biomedicines. However, the current paper had several flaws deterring acceptance. Overall, the manuscript could be reconsidered for publication in Pharmaceutics, provided a Resubmission was conducted. Please refer to the following comments:

1)      Please indicate the type of paper at the beginning of the template.

2)      A schematic illustration about the LbL structure of LNP should be displayed in the Introduction Section.

3)      The ATCC number of GL261 should be provided.

4)      A subsection about statistical analysis should be added at the end of the Methods Section.

5)      The detailed PDI values of the synthesized LNP should be provided.

6)      The current data was actually less sufficient to draw a conclusion. Please consider to perform further studies like the assay of loaded siRNA, the dose-dependent cytotoxicity towards GL261-Luc and the safety on normal cells.

7)      Although interesting, the rationale to develop 5-layer LbL was not clearly demonstrated. Was the number of layers strongly associated with the transfection efficiency? Why 5 layers was better than 3 layers?

8)      In the Discussion Section, reflections about industrialization translation could be supplemented.

Reviewer 3 Report

Comments and Suggestions for Authors

Dear author,

The paper entitled “Modulating the nature of ionizable lipid and number of layers in hyaluronan-decorated lipid nanoparticles for in vitro delivery of siRNA” has been intensively reviewed and evaluated. Although the present study was considered an interesting study, some points need to be revised. Hereby, I would like to present my suggestions and revisions.

Revision_1: line1_ Authors must specify whether it is an article, review, etc.

Revision_2: Authors must include data and results in the abstract

Revision_3: lines 59-60_ This definition needs a reference. below two references, which should also be commented on in the text: https://doi.org/10.3390/pharmaceutics14050942; https://doi.org/10.1083/jcb.116.4.1055

Revision_4: Session Synthesis and characterization of HA-LNPs_ Authors should add the name of the method of synthesis of the system.

Revision_5: Authors should also add in the materials and methods chapter how they performed significance (ANOVA)

Revision_6: The authors conducted a brief study. Could they expand with some in vitro biological studies? e.g., a cytotoxicity test?

Round 2

Reviewer 2 Report

Comments and Suggestions for Authors

The authors had well improved the manuscript. However, the reviewer suggested them to consider to conduct more in-depth investigations upon RNAi in their future studies.